# The Application of Artificial-Intelligence-Assisted Dental Age Assessment in Children with Growth Delay

**DOI:** 10.3390/jpm12071158

**Published:** 2022-07-17

**Authors:** Te-Ju Wu, Chia-Ling Tsai, Quan-Ze Gao, Yueh-Peng Chen, Chang-Fu Kuo, Ying-Hua Huang

**Affiliations:** 1Department of Craniofacial Orthodontics, Kaohsiung Chang Gung Memorial Hospital and Chang Gung University College of Medicine, Kaohsiung 833253, Taiwan; orthowilliam@gmail.com; 2Department of Pedodontics, Kaohsiung Chang Gung Memorial Hospital and Chang Gung University College of Medicine, Kaohsiung 833253, Taiwan; tsaicl612@gmail.com; 3Center for Artificial Intelligence in Medicine, Chang Gung Memorial Hospital, Taoyuan 333423, Taiwan; tirear@gmail.com (Q.-Z.G.); yuepengc@gmail.com (Y.-P.C.); 4Division of Rheumatology, Allergy and Immunology, Center for Artificial Intelligence in Medicine, Chang Gung Memorial Hospital, Linkou Medical Center, Taoyuan 333423, Taiwan; zandis@cgmh.org.tw; 5Department of Pediatrics, Kaohsiung Chang Gung Memorial Hospital and Chang Gung University College of Medicine, Kaohsiung 833253, Taiwan

**Keywords:** artificial intelligence, chronological age, convolutional neural network, dental age, Demirjian’s method, machine learning, population, Taiwanese, tooth development stage, Willems method

## Abstract

Background: This study aimed to reveal the efficacy of the artificial intelligence (AI)-assisted dental age (DA) assessment in identifying the characteristics of growth delay (GD) in children. Methods: The panoramic films matching the inclusion criteria were collected for the AI model training to establish the population-based DA standard. Subsequently, the DA of the validation dataset of the healthy children and the images of the GD children were assessed by both the conventional methods and the AI-assisted standards. The efficacy of all the studied modalities was compared by the paired sample *t*-test. Results: The AI-assisted standards can provide much more accurate chronological age (CA) predictions with mean errors of less than 0.05 years, while the traditional methods presented overestimated results in both genders. For the GD children, the convolutional neural network (CNN) revealed the delayed DA in GD children of both genders, while the machine learning models presented so only in the GD boys. Conclusion: The AI-assisted DA assessments help overcome the long-standing populational limitation observed in traditional methods. The image feature extraction of the CNN models provided the best efficacy to reveal the nature of delayed DA in GD children of both genders.

## 1. Introduction

Children’s growth condition is always a great concern to parents and practicians. The most common method of evaluating the growth rate is through serial body height measurements. During annual assessments, a growth delay (GD) is usually defined as a situation in which a child’s body height is below the third percentile [1,2]. The short stature of children with GD could be due to familial, genetic, endocrinological, and nutritional factors [3]. Conversely, delayed somatic development sometimes provides early diagnostic clues for underlying diseases [3], and children with GD are usually more vulnerable to psychosocial stress because of their reduced body heights [4]. In view of these profound influences, some studies have demonstrated the instrumentality of early treatment in correcting the body height to normal or near-normal levels [1,2]. Therefore, any remarkable clinical findings would play a crucial role in the early detection of GD.

As a biomarker, dental development has been widely discussed in forensic applications [5,6] and growth estimation [7,8]. Several studies have reported a delayed dental age (DA) in children with GD [2,4,9]. Both Kjellberg et al. [4] and Vallejo-Bolaños et al. [9] separately reported DA retardation in Sweden and Spanish children of short stature using the Finnish reference [10]. Meanwhile, Krekmanova et al. [2] also found that growth hormone substitution is helpful in accelerating delayed DA in a two-year follow-up conducted via Demirjian’s method (the D method), which was based on the French–Canadian reference [11]. However, the same assessment tool was unable to reproduce the same results in Indian children with GD [12].

Such efficacy differences are thought to be a result of the differences between the populations involved. The D method and Willem’s method (the W method) [13] are the two most common methods used for DA prediction because of their reliability and applicability. Both methods evaluate the DA by classifying the tooth bud formation and the tooth development staging (TDS) of the seven left mandibular permanent teeth and using conversion tables. In the D method, the users would firstly identify the TDS of the seven left mandibular permanent teeth and then find out the conversion scores according to the TDS classification of each studied tooth. Subsequently, the summation score of all seven teeth was used to reveal the corresponded estimated chronological age (CA) by checking the other conversion table. On the hand, the W method revealed the results by only one-time table checking, because each recorded TDS of each studied tooth turned out to correspond to years of age in the regression-modified table. As a result, the summation scores of all the seven studied teeth would directly point out the estimated CA. However, in a review, Jayakumar et al. reported a more-than-six-month overestimation of the CA by the D method and recommended that the estimation results should be explained with caution in varied populations [14]. Meanwhile, the W method provides the results with less overestimation but still with varying accuracy in different populations [15].

Recently, owing to the increased computation power and the evolving algorithm, artificial intelligence (AI)-assisted approaches have been introduced into the field of DA assessment. The reported results retrieved from either machine learning (ML) [16,17] or the convolutional neural network (CNN) [18] have shown their abilities to come up with more accurate estimations. For example, Antoine et al. compared different ML approaches with the D method and found more accurate results in ML estimations [16]. In addition, our previous study [17] also reported consistent agreements among preschool to adolescent groups in the Taiwanese population compared to standard population tables [19]. On the other hand, the CNN also presented comparable errors to those of traditional methods in the extended range of studied age groups [18]. All these findings prove that AI-assisted modalities were able to lift off the long-standing dilemma resulting from population diversities in DA assessment. It would be of great interest to determine how such improvements can help DA assessments in children with GD. Therefore, this study aimed to verify the efficacy of AI-assisted DA assessments, compared to the traditional methods, in children with GD.

## 2. Materials and Methods

In the present study, most of the studied images were collected from healthy children whose body heights were within normal ranges, and some were from children with GD whose body heights were below the third percentile in the growth chart. Considering the prevalence rate, the images of both healthy children and those with GD were collected using different strategies.

### 2.1. Data Collection

#### 2.1.1. The Images Collected from the Healthy Children

The studied panoramic films were retrospectively collected from the 2019–2020 database of Kaohsiung Chang Gung Memorial Hospital, Kaohsiung, Taiwan. The ages of the enrolled samples ranged from 3 to 18 years. To avoid possible misidentification, only those films presenting clear TDS were included. The medical records corresponding to the included samples were also carefully reviewed, and those with endocrine disorders, syndromic diseases, histories of early facial radiologic therapy, histories of major facial trauma, histories of previous orthodontic treatments, and tooth agenesis over the left mandibular segments were excluded. The sex ratio in each studied age group was approximately 1:1. The study protocol and procedures were approved by the Institutional Review Board of Kaohsiung Chang Gung Memorial Hospital (approval no. 201900844B0).

#### 2.1.2. The Images Collected from Children with GD

Considering its low prevalence, there were multiple search strategies for GD samples, including searching the Chang Gung Research Database from 2009 to 2019, chart review, and clinical referral. All the included samples had been diagnosed with GD without any evidence of systemic diseases, endocrine disorders, genetic abnormalities, and congenital craniofacial abnormalities. However, samples without clear records of less than 3% body height were also excluded, even if the diagnosis was GD. The ages of the studied samples also ranged from 3 to 18 years. All the included panoramic films had clear images without any deformation or major artifacts. The data collection procedures were approved by the Institutional Review Board of Kaohsiung Chang Gung Memorial Hospital (approval no. 201900864B0. 201900844B0C601).

#### 2.1.3. TDS Recording

The TDS of each enrolled sample was recorded according to the illustration and description of the D method [11]. Two senior dentists (TJW and CLC) underwent repeated training and practice sessions in order to reduce the possible bias from the manual identification. The reliability checks of the inter-examiner and intra-examiner validations were carefully arranged. A total of 50 panoramic films were randomly selected from enrolled samples for the reliability test. After the first identification, the disagreements between the two examiners were discussed to reach a consensus on TDS identification. Two weeks later, another randomly selected 50 samples were applied for another practice. On the other hand, for the intra-examiner validation, the same tested samples were re-checked again by the same examiner at a minimum interval of 2 weeks. It is until both inter-examiner and intra-examiner agreements attain near-perfect levels [20] that the overall TDA recording is initiated.

#### 2.1.4. The Development of AI-Assisted Modalities

To compare their efficacy with those of conventional methods, AI-assisted standards should be developed before the final validation. Eighty percent of the enrolled healthy samples were randomly selected as the training dataset to develop populational standards either by ML or by CNN models. Meanwhile, the remaining 20% of samples served as the validation dataset to compare the efficacy of traditional approaches to that of AI-assisted approaches.

#### 2.1.5. The Training of the ML Algorithm

According to our previous study [17], the Gaussian process regression (GPR) has the best performance among several tested algorithms in DA assessments. Therefore, we chose GPR for the efficacy comparison in the present study. The MATLAB’s Statistics and Machine Learning Toolbox (2021a release; MathWorks, Inc., Natick, MA, USA) was used for model developments. For the model training, the TDS of each tooth was transformed into numerical order. Values of 1 to 8 represented stages A to H, and absent tooth buds were registered as 0. The patient’s sex was another input parameter during the model training. To develop population standards, the training dataset retrieved from the initial 80% healthy samples was randomly divided for repeated training and validation by the principle of the five-fold cross-validation. Such a training sequence was repeated twenty times, and the model of the best performance was adopted for the following tests.

#### 2.1.6. The Training of the CNN Models

Different from the traditional methods using the TDS ranking as one of the input parameters, the CNN models in the present study took the image features amid the whole panoramic films for model development. All the input images were resized to a 704 × 704 resolution. Several data augmentation techniques, including rotation, blurring, motion blur, adding noise, etc., were applied. A CNN model based on the EfficientNet-B0 was adopted for training. The model was also subjected to the five-fold cross-validation, and a total of 400 epochs was applied before the final deployment.

#### 2.1.7. Statistics

To reach the optimal TDS identification consensus, Fleiss’s kappa [20] and the percentage of agreement for each tooth were used to assess the intra-examiner and inter-examiner reliability. The prediction error of each modality in both healthy children and those with GD of the validation dataset was calculated using the CA–DA. The efficacy levels of all the studied modalities were compared using the paired-sample *t*-test.

## 3. Results

In this study, a total of 2431 healthy children (1286 boys and 1145 girls) who matched the inclusion criteria were collected from the 2019–2020 image database of Kaohsiung Chang Gung Memorial Hospital, Kaohsiung, Taiwan. Among them, 2052 samples (1076 boys and 976 girls) constituted the training dataset with a nearly-balanced sex ratio in each age group, and the remaining 379 samples (210 boys and 169 girls) were used as the validation dataset. On the other hand, 99 children with GD (54 boys and 45 girls) were identified and enrolled in this study through amplified search strategies.

To validate the efficacy among studied modalities, a reliable TDS recording performance played an important role. In the present study, both the kappa value of the inter-examiner and intra-examiner agreements reached the almost perfect level of agreement at a Fleiss kappa of 0.802 and 0.825 individually. Both measurements had a *p*-value of <0.001, which showed that the agreements were identical to those that could be due to chance.

After using inputs from healthy children as the training dataset, the ML and CNN assisted the development of population standards. AI-assisted standards were then used to compare the efficacy of the traditional methods by testing the validation dataset and the data of children with GD. According to the results, both AI-facilitated models presented more accurate CA prediction results than the two conventional methods in both sexes. The mean errors of the AI-facilitated methods were no more than 0.05 years, while both traditional methods tended to show greater overestimated results of statistical significance (Table 1).

For children with GD, the D method tends to yield overestimated results with statistical significance in both sexes, while the W method showed no significant difference. On the other hand, the CNN model revealed underestimated results with statistical significance in both sexes, while the ML model yielded such results only in boys with GD (Table 2).

The identified features revealed by CNN models in both healthy and GD children were compared at the same age. The “delayed” pattern was observed in the GD children and the possible rules of CNN-extracted patterns were discussed (Figure 1).

## 4. Discussion

Children’s growth conditions are always a great concern to parents and pediatricians. It has been reported that children with GD tend to have DA delay [2,4,9]; however, controversial results were also reported among different populations [12]. The population difference is such a long-standing limitation that influences the clinical application of the DA assessment [14,15,21]. Recently, it has been reported that such a limitation could be overcome using AI techniques [16,17]. Therefore, the present study aimed to reveal the efficacy of the AI-assisted DA assessment in identifying the features of children with GD.

In this study, the training datasets were collected from children with normal growth rates and were used to develop the Taiwanese DA standard using AI techniques. According to the validation results, AI-assisted methods can produce more accurate CA estimations, while both traditional methods yielded overestimated results in Taiwanese children (Table 1). This finding is in line with those of the previous studies [16,17,18]. Subsequently, the developed AI-assisted standards were used to verify the DA characters in children with GD. According to the results, although to a lesser extent, the D method still revealed overestimated results in children with GD and the W method revealed nonsignificant differences in both sexes. On the other hand, the CNN revealed a DA delay in GD children of both sexes, while the ML yielded this result only in boys with GD (Table II). In the CNN model, girls with GD tended to have a >0.5-year DA delay, and greater delays that were close to one year were presented in GD boys. Briefly, the nature of the DA delay was more easily revealed by AI-assisted DA assessments in children with GD.

To develop DA assessment standards, the TDS and image features were separately used as input parameters in addition to children’s sexes. Firstly, among all studied modalities, both traditional methods and the ML model used the TDS and sex as parameters to deduce special laws of induction with or without conversion tables for the CA prediction. The results showed that the ML model has much higher CA prediction accuracy than the two traditional methods with the same parameters. The increased accuracy of the ML model also implied the importance of a population-based standard when DA-related issues were discussed. However, we also noticed that the AI-assisted standard was sufficiently empowered to reveal the nature of DA delay in GD children of both sexes under the TDS-based training (Table 2).

On the other hand, the CNN model was the only modality capable of presenting the DA delay in both male and female children with GD. Different from the above-mentioned methods, the CNN model used the whole panoramic image as the input during model training. The image features of panoramic films were identified and extracted while the CA was set as an output standard. This concept has been tested in a previous study [18]. Kim et al. used another CNN model to test the CA prediction accuracy in samples of the prepuberty, puberty, and postpuberty periods without taking sex as a parameter. According to their results, the mean absolute errors were 0.826 and 1.229 years before and during puberty, respectively [18]. In our case, the mean absolute error was 0.421 years in all samples without sex differences. We believe such improvements could result from the recently increased computing power and algorithm optimization in the AI field.

For a long time, researchers have kept exploring the possible clues associated with children’s growth and development. As a routine dental examining tool, we found that the panoramic film could provide supplemental information about growth assessments. According to the feature identification of the CA prediction process, the specific area and sequences seemed they could be traced and were worthy of our attention (Figure 1). In the youngest group of samples, feature identification was initiated from the cervical region of the primary incisors followed by the coronal region of permanent incisors, the left mandibular primary and permanent molars, the right mandibular primary and permanent molars, the coronal region of the left mandibular premolars, the coronal region of the right mandibular premolars, and the mandibular structures.

Interestingly, several features or rules have been once studied individually in previous studies. For example, the left-side dominant rule was used in the D and W method [11,13]. Additionally, the regions of interest located at the cervical and coronal portions of teeth have also raised several topics associated with age prediction. First, could the mineral density of the dental structures be an indicator of the growth condition? Amelogenesis is a multifactorial process, and enamel defects are frequently observed in many genetic and/or systemic disorders [22]. A previous case-control study reported the nonsignificant association between molar-incisor hypomineralization and the DA estimation [23]. Furthermore, another study also revealed a nonsignificant correlation between molar-incisor hypomineralization and short stature [24]. Therefore, it seems the mineral density of a tooth does not play a role in DA estimation. Secondly, the application of the linear pulp/tooth ratio as an indicator of age estimation has also been discussed [25,26,27]. Although a negative correlation was found between the linear pulp/tooth ratio and age [25], the reported error remained huge (<4 years) [26]. However, one recent CBCT study reported a strong correlation between the volumetric pulp/tooth ratio and the estimated age [28]. The detailed connection and interpretation of AI-extracted features would be of great interest in future studies.

Although the present study has proven the potential of AI-assisted DA assessments to reveal the GD nature of affected children, it still had certain limitations. First, somatic and craniofacial growth are controlled by multifactorial mechanisms. The effects of the other environmental, socioeconomical, and nutritional factors need to be better clarified in future studies to ensure the higher reliability of DA assessments. Second, the ages of the study participants were limited and did not extend to adulthood. It would be worthy of notification if the various dental interventions, including prosthetic treatments, root canal filling, and tooth extraction, influence AI performances. Lastly, this was a descriptive study; therefore, more clinical observations and prospective studies are expected to provide accurate predictions of long-term changes.

## 5. Conclusions

In the present study, we compared the efficacy of AI-assisted DA assessments with that of two conventional methods in both healthy children and those with GD. We found much higher CA prediction accuracy after establishing the AI-assisted standard that successfully overcame the population-related limitations observed with traditional methods. According to the results, the delayed DA could be only observed in GD boys in the ML-assisted evaluation. On the other hand, the CNN model, which extracts image features from whole panoramic films, has the best efficacy in revealing the nature of delayed DA in both boys and girls with GD. In addition to the TDS scoring, the possible alternative features relating with the CA prediction or the developments of the craniofacial structures were also discussed. However, the results of the present study should be applied cautiously because of the limitations from the limited studied parameters, the uncertain influences of image artifacts resulted from dental procedures, and the lack of long-term results.

## Figures and Tables

**Figure 1 jpm-12-01158-f001:**
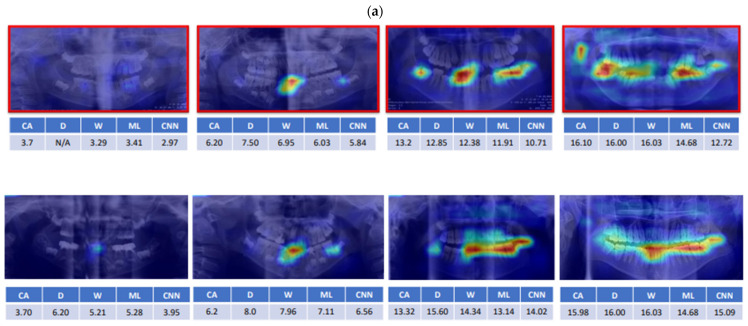
The feature identification of the convolutional neural network (CNN) models in samples. The comparison of the identified features by the CNN models between the healthy children and children with growth delay (GD) at the same age ((**a**) boys, (**b**) girls). The images of GD children are framed in red color. The coloring indicated the order of the increased intensity of attention from blue, green, yellow to red. The area and intensity of the feature identification in the GD children are relatively delayed compared to that of healthy ones. Abbreviations: chronological age (CA), D method (D), W method (W), machine learning (ML), convolutional neural network (CNN).

**Table 1 jpm-12-01158-t001:** The performance of studied methods in predicting CA of healthy children.

	**Girls**				
**Prediction Error**	**N**	**Mean (Age)**	**Std. Deviation (Age)**	**Std. Error Mean (Age)**	***p* Value**
CA-D method	169	−0.818	0.852	0.066	0.000 *
CA-W method	169	−0.279	0.792	0.061	0.000 *
CA-ML	169	0.039	0.736	0.057	0.488
CA-CNN	169	0.014	0.718	0.055	0.793
	**Boys**				
**Prediction Error**	**N**	**Mean (Age)**	**Std. Deviation (Age)**	**Std. Error Mean (Age)**	***p* Value**
CA-D method	210	−0.926	1.005	0.069	0.000 *
CA-W method	210	−0.468	0.917	0.063	0.000 *
CA-ML	210	−0.050	0.770	0.053	0.345
CA-CNN	210	0.007	0.637	0.044	0.874

* *p* < 0.05. Abbreviations: chronological age (CA), number of samples (N), D method (D method), W method (W method), machine learning (ML), convolutional neural network (CNN).

**Table 2 jpm-12-01158-t002:** The performance of studied methods in predicting CA of GD children.

	**Girls**				
**Prediction Error**	**N**	**Mean (Age)**	**Std. Deviation (Age)**	**Std. Error Mean (Age)**	***p* Value**
CA-D method	45	−0.694	1.508	0.225	0.003 *
CA-W method	45	−0.244	1.503	0.224	0.283
CA-ML	45	0.252	1.301	0.194	0.201
CA-CNN	45	0.466	1.044	0.156	0.005 *
	**Boys**				
**Prediction Error**	**N**	**Mean (Age)**	**Std. Deviation (Age)**	**Std. Error Mean (Age)**	***p* Value**
CA-D method	54	-0.318	1.051	0.147	0.036 *
CA-W method	54	-0.048	1.090	0.148	0.746
CA-ML	54	0.497	0.971	0.133	0.000 *
CA-CNN	54	0.902	1.158	0.158	0.000 *

* *p* < 0.05. Abbreviations: chronological age (CA), growth delay (GD), number of samples (N), D method (D method), W method (W method), machine learning (ML), convolutional neural network (CNN).

## Data Availability

Data are available on request but may be restricted for reasons such as privacy or ethical concerns.

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
