# Peer review of "The Application of Artificial-Intelligence-Assisted Dental Age Assessment in Children with Growth Delay"

_jpm, 2022, doi:10.3390/jpm12071158_

Round 1

Reviewer 1 Report

How can the accuracy or correctness of this analysis be verified?

The time allotted for this study is too short.

Changes and errors cannot be evaluated in one year.

The images are not explicit enough. These are not relevant enough to give readers an overview of the study.

The results mentioned in the conclusions are not sufficiently supported by the whole study.

What is the advantage and the novelty?

Reviewer 2 Report

The manuscript “The application of artificial-intelligence-assisted dental age assessment in children with growth delay” is a study that compares the efficacy of identification of grow delay through the dental age between the traditional methods and the artificial intelligence methods. The paper falls within the scope of the journal. English language and style are fine and minor spell check is required.

Some minor flaws as follows:

       Title: the title expresses clearly what the manuscript is about. No chances required.

       Abstract: the abstract is structured in according to editorial standards; the aim of the study is specified.

       Introduction: the introduction summarizes the current state of the topic and the aim of the study is clear. For better understanding of the text the cited “Demirjian’s method” and “Willem’s method should be clearly explained.

       Materials and methods: this part is complete and comprehensive. The inclusion and exclusion criteria of the study are well defined. The “TDS Recording” section should be explained more clearly.

       Results: It is not specified where patients with grow delay were recruited. Furthermore, it is unusual to include references in the results; in this sense, the section between lines 161-164, page 4, should be transposed in the previous section of material and methods.

    Discussion: the section is presented thoroughly and in detail; images are clear and easy to read, coherent to the text.

Conclusion: the conclusions are too concise. The section should be more detailed, and the limits of the current research should be explained. 

Reviewer 3 Report

The aim of this study was to test the effectiveness of AI-assisted dental age assessments, compared to traditional methods, in children with growth retardation. This article is timely and well written. Just one minor point that is reported below.

I suggest moving to the results section the chapter where the authors have outlined the comparison of characteristics identified by the CNN models between healthy and stunted children at the same age.

Round 2

Reviewer 1 Report

Ok